# Preparation of Needleless Electrospinning Polyvinyl Alcohol/Water-Soluble Chitosan Nanofibrous Membranes: Antibacterial Property and Filter Efficiency

**DOI:** 10.3390/polym14051054

**Published:** 2022-03-07

**Authors:** Ching-Wen Lou, Meng-Chen Lin, Chen-Hung Huang, Mei-Feng Lai, Bing-Chiuan Shiu, Jia-Horng Lin

**Affiliations:** 1Fujian Key Laboratory of Novel Functional Fibers and Materials, Minjiang University, Fuzhou 350108, China; cwlou@asia.edu.tw; 2Advanced Medical Care and Protection Technology Research Center, College of Textile and Clothing, Qingdao University, Qingdao 266071, China; 3Department of Bioinformatics and Medical Engineering, Asia University, Taichung City 413305, Taiwan; 4Department of Medical Research, China Medical University Hospital, China Medical University, Taichung City 404333, Taiwan; 5Laboratory of Fiber Application and Manufacturing, Department of Fiber and Composite Materials, Feng Chia University, Taichung 407102, Taiwan; lai3630@gmail.com; 6Department of Aerospace and Systems Engineering, Feng Chia University, Taichung City 407102, Taiwan; 7College of Material and Chemical Engineering, Minjiang University, Fuzhou 350108, China; toyysbk2@yahoo.com.tw; 8Advanced Medical Care and Protection Technology Research Center, Department of Fiber and Composite Materials, Feng Chia University, Taichung City 407102, Taiwan; 9School of Chinese Medicine, China Medical University, Taichung City 404333, Taiwan

**Keywords:** polyvinyl alcohol, chitosan, nanofibers, filtration efficiency, antibacterial property

## Abstract

Electrospinning is an efficient method of producing nanofibers out of polymers that shows a great potential for the filtration territory. Featuring water-soluble chitosan (WS-CS), a low-pollution process and a self-made needleless machine, PVA/WS-CS nanofibrous membranes were prepared and evaluated for nanofiber diameter, bacteriostatic property, filtration efficiency, pressure drop, and quality factor. Test results indicate that the minimal fiber diameter was 216.58 ± 58.15 nm. Regardless of the WS-CS concentration, all of the PVA/WS-CS nanofibrous membranes attained a high porosity and a high water vapor transmission rate (WVTR), with a pore size of 12.06–22.48 nm. Moreover, the membranes also exhibit bacteriostatic efficacy against *Staphylococcus aureus*, an optimal quality factor of 0.0825 Pa^−1^, and a filtration efficiency as high as 97.0%, that is 72.5% higher than that of common masks.

## 1. Introduction

The removal of solids and particles from the air or liquids with filters is required by production processes or other types of work in the manufacturing industry. For example, drug, semiconductor, medical treatment materials, biotechnology, drinking water, and even air conditioning industries employ filtration technology to assure a pollution-free work environment and non-polluting products [1,2,3,4,5,6]. Alternative materials for filters with corresponding purposes include the following: polypropylene (PP), polyethylene (PE), and polyurethane (PU) for air filters [7,8]; Nomex^®^, glass fibers, and Teflon^®^ for hot gas filtration [9]; and activated carbon [10,11], ion exchange resin [12,13], and ceramic membranes [14,15,16,17] for drinking water and sewage disposal. Notably, a majority of filter materials are non-degradable, and the waste filters become a new environmental issue. Natural or endurable materials are increasingly popular filter materials, encouraging reuse and durability for sustainable development with the least harm to both people and nature [18,19,20,21,22,23,24]. However, industrial development has caused pollutions that jeopardize the environment and human health constantly, especially particulate pollutants that have become a global crisis. A great number of studies have proved that a long-term exposure to a polluted environment inflicts the human body with respiratory system disorder, apoplexy, heart disease, and cancer [24,25,26,27,28,29]. Therefore, people start seeking suitable and effective filters in order to protect public health.

In recent years, filters are pervasively made using the melt blowing or spunbond techniques, and thus nanofibers have drawn much attention from diverse fields. Owing to the unique advantages, e.g., a small fiber diameter, a large specific surface area, and a low pressure drop [30,31,32,33], nanofibers are a feasible material for filters [24,34,35,36,37]. Electrospinning is effective for transforming materials into ultrafine fibers, and the related techniques are similar to other techniques in current industry, producing micro-sized fibers, including melt spinning, wet spinning, dry spinning, and gel spinning [38]. By contrast, the electrospinning technology draws the viscoelastic polymer fluid into a jet via the repulsion from electric charges of fluid surface against a high voltage electric field, thereby spins the nanofibers out of the jet [39]. In addition, electrospinning is more capable of producing ultrafine fibers than conventional spinning, that only employs a mechanical force to spin fibers.

Electrospinning evolved from the traditional needle electrospinning to the coaxial nozzle, enriching the functions of nanofibers but failing in mass production [40,41,42]. The successive multi-needle electrospinning fulfills the demands of mass production, yet it still needs to overcome difficulties, e.g., needle clogging and electric field interference [43,44]. The current needleless electrospinning technique improves on the aforementioned difficulties, and has been commonly used for effective mass production of nanofibers. To increase the yield of nanofibers, many scholars have proposed diverse jet ends so far, such as cylinder spinneret [45], spiral coil spinneret [46], disk spinneret [47], magnetic-field-assisted multi-spikes electrospinning [48], and porous tubular surface electrospinning [49].

As a hydrophilic polymer, polyvinyl alcohol (PVA) is a toxin-free and efficient film-forming, solvent-resistant, water-soluble, gas-barrier, and biocompatible material, so PVA is commonly used as an adhesive for fabrics as well as an emulsifier for biomaterial and cosmetics [50,51,52,53,54,55]. Besides, chitosan (CS) is a natural polymer that has the bacteriostatic property, chelation, biocompatibility, biodegradability, and avirulent feature. CS has received widespread acceptance in many biomedical applications, such as wound dressings, biologic scaffolds, drug delivery, and metal ion adsorption [56,57,58,59,60,61,62]. Nonetheless, chitosan is insoluble and usually combined with an acid solvent, which in turn leaves the resulting products with acid residue that hampers cell growth [56,58,63,64,65,66]. The presence of water-soluble chitosan can address the problem, because organisms can absorb chitosan efficiently. Meanwhile, the structure of soluble chitosan also exhibits better affinity to molecules, which in turn shows a positive influence on the antimicrobial effect [67,68,69,70,71].

There are numerous studies reporting the results of a combination of electrospinning, PVA, and chitosan (CS). Paipitak et al. successfully produced PVA/CS nanofibers. Based on the functional groups of PVA and CS, observed with Fourier transform infrared spectroscopy (FT-IR), a high concentration of PVA/CS blends caused a high viscosity that stabilized the formation of fibers [72]. Similarly, Elmira et al. found that a greater CS ratio helped generate a sleek surface of CS/SS/PVA nanofibers, and CS also worked well in terms of bacteriostatic efficacy against *E**. coli* [73]. Wang et al. produced PVA/CS membranes and found that an intermolecular hydrogen bond occurred between PVA and CS. With a CS concentration being 30%, the membranes had a filtration efficiency of 95.59% and a pressure drop of 633.5 Pa, along with antibacterial efficacy against both *E. coli* and *S. aureus* [74]. Similarly, Zou et al. developed PVA/CS nanofibrous membranes. When OH-30 nanoparticles were incorporated, the membranes could facilitate wound healing and demonstrate the antibacterial efficacy against *E. coli* and *S. aureus* concurrently [75]. The hydrophilicity of PVA provides PVA/CS nanofibrous membranes with degradation as required. When used for metal ion adsorption filtration, water filtration, special gas filtration, a high temperature condition, and the extended drug release of wound dressings, PVA/CS membranes demand a specified processing, e.g., thermal cross-linking, hydrogen bond cross-linking, or solvent cross-linking, thereby attaining more general applications. However, the specified processing is usually accompanied with toxic solvents [76,77,78,79,80,81,82].

According to a large majority the literature, when used in metallic filters, wound dressings, and food packaging, scholars prefer general CS membranes over water-soluble chitosan nanofibers. Meanwhile, scholars explored the types or antibacterial performances of CS nanofibrous membranes rather than their filtration efficiency. Promoting natural or low-pollution materials has become a trend in recent years, so this study proposes one kind of degradable one-off nanofibrous membrane serving as the air filter material. Different concentrations of WS-CS were used as the bacteriostatic agent, comprising PVA/WS-CS nanofibrous membranes, and as such avoids the problem regarding the residue of acetic acid. Finally, the morphology, water vapor transmission rate, bacteriostatic efficacy, and filterability of PVA/WS-CS nanofibrous membranes were evaluated accordingly.

## 2. Materials and Methods

### 2.1. Materials

Polyvinyl alcohol (PVA) powders (Sigma-Aldrich, St. Louis, MO, USA) have a molecular weight of 89,000–98,000 Da. Water-soluble chitosan (WS-CS) powders (Charming and Beauty Co., Taipei, Taiwan) have a molecular weight of 30,000 Da and a deacetylation of 85%.

### 2.2. Preparation of Nanofibers

Figure 1 illustrates the diagram how nanofibrous membranes are made with a self-made needleless machine. PVA powders were added to deionized water, forming a PVA solution with a specific concentration being 10 wt%, and likewise WS-CS powders were added to deionized water, forming WS-CS solutions with concentrations of 5, 10, and 15 wt%. Next, with a specified total mixture volume of 50 mL, PVA and different WS-CS solutions were mixed at volume ratios of 100/0, 80/20, and 60/40, and were agitated well for 1 h. A PVA solution and different PVA/WS-CS mixtures separately underwent the electrospinning process with parameters of voltage 50 kV and the distance between the nozzle and the collector plate of 15 cm, thereby producing PVA/WS-CS electrospinning nanofibrous membranes.

### 2.3. Scanning Electron Microscopy (SEM) Observation and Measurement

PVA/WS-CS nanofibrous membranes made of different blending ratios were coated with a thin layer of gold, and then they were mounted in a scanning electron microscope (S-4800, Hitachi Ltd., Tokyo, Japan) for observation and photographing. According to the SEM images, the diameters of the nanofibers were measured employing the Image-Pro Plus 6.2 (Media Cybernetics, Inc., Rockville, MD, USA).

### 2.4. Viscosity and Conductivity Tests

The viscosity and conductivity of PVA/WS-CS mixtures at different ratios were measured using a rotational viscometer (Viscobasic+L, Fungilab, Barcelona, Spain) and a pH/conductivity meter (EC500, Extech Instruments, Nashua, NH, USA), respectively.

### 2.5. Porosity and Pore Size Tests

The porosity and pore size of PVA/WS-CS nanofibrous membranes were measured using the BET analyzers (ASAP 2000, Micromeritics Instrument Corp., Norcross, GA, USA) with all the samples being vacuum dried for 24 h in advance.

### 2.6. Water Vapor Transmission Rate (WVTR) Test

As specified in ASTM E96, the water vapor transmission rate (WVTR) of PVA/WS-CS nanofibrous membranes was measured in an airtight test box at a temperature of 25 °C and relative humidity of 30–35%. The initial weight (*W*_0_) of sample bottle was weighed using a balance, after which the sample bottle was mounted in the measurement case. After 24 h, the sample bottle was weighed once again (*W_t_*) and then WVTR was computed with the following equation:(1)$$\mathrm{WVTR}=\frac{\left(W_{0}-W_{t}\right)}{\left(A\times t\right)}\times 100\;$$
where *W*_0_ is the initial weight (g) of sample bottle (including a glass bottle, water, and a nanofibrous membrane), *W_t_* is the weight (g) of sample bottle after a 24 h WVTR test, *A* is the test area (m^2^), and *t* is the volatilization time (h) for the water vapor.

### 2.7. Bacteriostatic Assay

As specified in the JIS1902-2002 test standard, *Staphylococcus aureus* (*S. aureus*) was used for bacteriostatic assay. For the starter, 100 μL of *S. aureus* was dripped and then smeared evenly over a solid agar. Next, a perforator was used to make PVA/WS-CS nanofibrous membranes into circular samples with a 6 mm diameter. The membranes then covered the solid agars that were individually smeared with *S. aureus*. The test was conducted for 24 h, after which the inhibition zones surrounding the membranes were observed, thereby examining the bacteriostatic efficacy.

### 2.8. Fourier Transform Infrared Spectroscopy (FTIR)

FTIR was conducted to analyze the functional groups of PVA/WS-CS nanofibrous membranes using a Fourier transform infrared spectrometer (Spectrum Two, PerkinElmer Inc., Waltham, MA, USA). The spectra were in a scanning range of 400–4000 cm^−^^1^.

### 2.9. Filterability Measurement

The filterability of different PVA/WS-CS nanofibrous membranes was measured using an electrical low-pressure impactor (ELPI^TM^, Dekati Ltd., Kangasala, Finland). The test was conducted with the parameters as follows. The flow rate of gas was 85 ± 4 L/min, while the particle concentration of sodium chloride (NaCl) was 200 mg/m^3^. Serving as a filter, a nanofibrous membrane that was positioned horizontally started retaining NaCl particles with a 10-min suction by a downward gas flow. Concurrently, a small proportion of NaCl particles passed through the nanofibrous membrane and entered the case of the ELPI. Afterwards, the NaCl concentration in the ELPI was measured and defined as the filterability of the membrane. Next, the pressure difference between two sides (upper and lower sides) of the membrane was measured using a micromanometer (Models PVM 610, Airflow Measurements Ltd., Bolton, UK), after which the protection efficiency (PE) of PVA/WS-CS nanofibrous membranes was computed with the following formula:(2)$$\mathrm{PE}=\frac{C_{0}-C_{i}}{C_{0}}\times 100\;\%$$
where PE is the protection efficiency (%), C_0_ is the NaCl concentration (mg/m^3^) before filtration, and C_i_ is the NaCl concentration (mg/m^3^) after filtration.

## 3. Results and Discussion

### 3.1. Morphology and Diameter of Nanofibers

Figure 2 shows the morphology of nanofibrous membranes as related to the voltage. In the pilot experiment, a pure PVA solution with a concentration being 10 wt% was used. When the voltage increased from 30 kV to 50 kV, finer nanofibers were obtained, but when it increased to 70 kV, the nanofibers were accompanied with tremendous beads. The electric field force was employed to draft the polymer solution into nanofibers during the electrospinning process. A lower voltage caused an insufficient electric field force, and thus nanofibers could not be drafted into a finer scale and the yielded nanofibers appeared to have uneven diameters. With a rise in the voltage, polymer solutions could be drafted into finer nanofibers, which means the yielded nanofibers were finer, with an even diameter. An excessive voltage gave rise to an over-powerful electric field force, which triggered excessive jets, such that electrospinning could not operate stably. Inevitably, the nanofibers were unevenly produced and became bead-shaped [44,83], which determined that the voltage was 50 kV in the subsequent experiment.

Figure 3 shows the SEM images and fiber diameter diagrams of PVA/WS-CS nanofibrous membranes as related to the PVA/WS-CS blending ratio and the WS-CS concentration. With a blending ratio of 80/20, as in Figure 3a–c, the yielded nanofiber diameter was smaller than those made with a blending ratio of 60/40, as in Figure 3d–f. The observation suggests that the diameter of the nanofibers has an increasing trend when the specific weight of water-soluble chitosan (WS-CS) is increased. Because WS-CS and PVA molecular chains have an interaction force, the mixtures obtained a greater viscosity [72,84]. Usually, when the molecular chains of polymers are entangled and then reach a critical point, the mixture can be successfully drawn by an electric field force to form nanofibers. With a low viscosity, the mixture falls short of molecular chain entanglement, and is prone to the presence of electro-spraying. With a high viscosity, the mixture may be hampered by the high cohesiveness of the solution, which in turn causes an unstable jet that may fail to form electrospinning nanofibers [85]. In addition, the diameter of nanofibers is also dependent on the viscosity. As found in a previous study, a solution with a low viscosity was prone to form finer nanofibers, whereas a solution with a high viscosity might generate thicker nanofibers. In the electrospinning process, a polymer solution generates counterforce against the electric field force, which means that it becomes more difficult for a solution with a higher viscosity to be drafted by a high voltage, and the resulting nanofibers were finer [30,32].

Additionally, regardless of whether the PVA/WS-CS blending ratio was 100/0, 80/20, or 60/40, when WS-CS concentration increased, the diameter of nanofibers became smaller accordingly. Specifically, the 60/40-15 group obtained the smallest diameter of nanofibers of 216.58 ± 58.15 nm; interestingly, this group happened to have the highest viscosity, as shown in Table 1. The results are ascribed to the high electrical conductivity of WS-CS, which provided the charged polymer solution with a higher charge density, which in turn created finer nanofibers [61,72,85].

### 3.2. Porosity, Pore Size, and Water Vapor Transmission Rate (WVTR)

Table 2 summarizes the porosity, pore size, and WVTR of PVA/WS-CS nanofibrous membranes. Firstly, PVA/WS-CS nanofibrous membranes show a greater porosity than commercially available masks, which highlights a high porosity of the proposed membranes [86,87]. Next, the pore size of the membranes (12.06–22.48 nm) is smaller than that of the mask (2824.07 nm). A smaller pore size means that the nanofibrous membranes can intercept smaller particulates; therefore, they exhibit better filtration efficiency in the subsequent test. As for the WVTR, the masks with a largest pore size that facilitated volatilization of water vapor demonstrated the maximal WVTR among all groups. By contrast, PVA/WS-CS nanofibrous membranes showed a greater WVTR when at a blending ratio of 80/20 than at a blending ratio of 60/40. Because PVA is a hydrophilic material containing a great number of hydroxyl group, PVA/WS-CS nanofibrous membranes attract more water vapor to pass through [55,88]. In particular, the 60/40-05 group demonstrates the maximal WVTR because the resulting nanofibers were thicker and bead-shaped, which increased the pore size among nanofibers, as well as the permeability of the membranes [89,90,91].

### 3.3. Bacteriostatic Property

Figure 4 shows the bacteriostatic zone of PVA/WS-CS nanofibrous membranes as related to the PVA/WS-CS blending ratio (80/20 and 60/40) and the WS-CS concentration (5, 10, and 15 wt%). The bacteriostatic zone increased when WS-CS concentration rose, which suggests that PVA/WS-CS nanofibrous membranes demonstrate a strengthened bacteriostatic property. The test result is in conformity with the finding of Santiago-Castillo et al., which substantiates that PVA/CS nanofibrous membranes were antibacterial [92]. Similarly, in the study by Nokhasteh et al., alkali treatment was used to improve the water solubility of chitosan, and they also found that PVA/CS nanofibrous membranes were antibacterial [84]. By contrast, Yousefi et al. reported that the presence of CS did not provide samples with antibacterial efficacy against *Staphylococcus aureus* (*S. aureus*), which was ascribed to a low CS concentration [93]. *S. aureus* is Gram-positive bacteria, and like most bacteria, *S. aureus* contains cell walls carrying negative electrification. As a cationic polymer, WS-CS consists of amino groups. When amino groups are in contact with cell walls, the interaction makes the structure of cell walls unstable, which in turn makes the matter from the interior exposed. In this case, amino groups damage the cell walls and then enter the cells to synthesize with the DNA and RNA, hindering mRNA and protein from transcription and translation, and, as such, causes the death of cells eventually [94,95].

### 3.4. FTIR Analysis

Figure 5 shows FTIR spectra of PVA/WS-CS nanofibrous membranes as related to the PVA/WS-CS blending ratios. Pure PVA nanofibrous membranes (the control group) shows the O-H stretching vibration at 3316 cm^−1^, the stretching C-H of alkyl groups at 2944 cm^−1^, the stretching vibration of C-OH and C-C separately at 1090 cm^−1^ and 850 cm^−1^, respectively, the -CH_2_ stretching at 1425 cm^−1^ and 1250 cm^−1^, respectively, and the -C=O stretching peaks at 1718 cm^−1^. By contrast, PVA/WS-CS nanofibrous membranes consist of O-H stretching vibration at 3308 cm^−1^, Amide I and Amide II characteristic peak at corresponding 1639 cm^−1^ and 1531 cm^−1^, respectively, and C-H stretching characteristic peaks at 2869 cm^−1^. The FTIR spectra indicate that PVA/WS-CS nanofibrous membranes are composed of the characteristic peaks belonging to both PVA and WS-CS, which is in conformity with the findings in the bacteriostatic property test. Namely, the bacteriostatic property of the membranes is mainly attributed to WS-CS that contains amino groups. Meanwhile, the FTIR spectra also substantiated that PVA/WS-CS nanofibrous membranes were composed of WS-CS functional groups [50,56,96,97,98,99,100,101,102,103].

### 3.5. Filtration Efficiency

Figure 6 shows the filtration efficiency and pressure drop of PVA/WS-CS nanofibrous membranes. Notably, all PVA/WS-CS nanofibrous membranes exhibit a filtration efficiency that is higher than 90% and the maximal filtration efficiency achieves 97%, which is considerably greater than the filtration performance of masks. Common filtration mechanisms include the inertial impaction, direct interception, diffusional interception, sieving, and gravitational settling that intercept particles from passing through. In this study, mechanical filtration was adopted; therefore, nanofibrous membranes with pore sizes much smaller than common filters are able to block particles with sizes smaller than 2.5 µm [2,104]. Moreover, the pressure drops of the PVA/WS-CS nanofibrous membranes were between 41 Pa and 44 Pa without significant differences. Although this was marginally higher than that of the mask (40 Pa), it is still inferior to that of N95 (48 Pa). Specifically, the 60/40-5 group exhibits a rather smaller pressure drop because the constituent nanofibers were thicker, indicating that larger pores among stacked nanofibers facilitate the air going past. Moreover, Zhu et al. also found a similar filtration efficiency as they produced CS-PVA@SiO_2_ NPs nanofibrous membranes via electrospinning and the membranes acquired a filtration efficiency that was greater than 96% [105]. Cui et al. incorporated PVA nanofibers with masks, and the products achieved a 99% filtration efficiency against particles that was smaller than PM1.0 [106]. The findings of previous studies prove that nanofibrous membranes are a powerful and effective material for filtering and purifying the air. Generally speaking, the service life of filters is correlated with the feature of pressure drop, and a greater pressure drop means that the air passes through the filter with more difficulty. Subsequently, the filter will intercept increasingly accumulated particles with time, forming a barrier against the air flux and compromising the service life of filter [107,108,109].

Figure 7 shows the filtration efficiency of PVA/WS-CS nanofibrous membranes as related to the particle size. Regardless of the particle size, the membranes exhibit excellent filtration efficiency, especially particles smaller than 2.5 µm. The filterability was as high as 93.1%, whereas masks only intercept 72.5% in the same condition.

Lastly, Figure 8 shows the quality factor of PVA/WS-CS nanofibrous membranes with a correlation equation as $Q_{f}=\frac{-\ln\left(1-E\right)}{\Delta P}$, where Q_f_ is the quality factor; E is the filtration efficiency; and ∆P is the pressure drop. According to Figure 8, PVA/WS-CS nanofibrous membranes have a much greater quality factor than masks, and despite the highest pressure drop, the 60/40-15 group still retains the optimal quality factor. As far as filter evaluation is concerned, the pressure drop is not the only standard to determine the quality, yet there are a great number of influential factors, e.g., thickness, pore size, and specific area [2,104].

## 4. Conclusions

With a low-pollution process and a self-made needleless machine, PVA/WS-CS nanofibrous membranes were successfully made with water-soluble chitosan and PVA in this study. The test results confirm that the proposed 80/20 and 60/40 blending ratio of PVA to WS-CS can be electrospun into PVA/WS-CS nanofibers, and the 60/40-15 group possesses the smallest diameter of nanofibers. Furthermore, all the proposed PVA/WS-CS nanofibrous membranes have an average pore size of 12.06 nm–22.48 nm, a high porosity, and a high water vapor transmission rate (WVTR), in addition to the excellent bacteriostatic property and filtration efficiency, which blocks particles that are smaller than 2.5 µm. The optimal filtration efficiency is between 93.1% and 97%, which is higher than that of commercially available masks (72.5%), while an optimal pressure drop was found to be between 41 Pa and 44 Pa, suggesting that PVA/WS-CS nanofibrous membranes outperform masks in terms of excellent quality factor and are a qualified candidate for use in the filtration field.

## Figures and Tables

**Figure 1 polymers-14-01054-f001:**
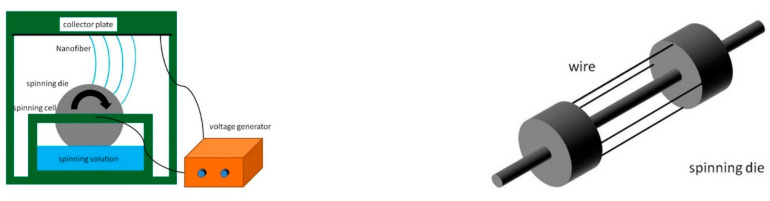
Configuration of the electrospinning assembly.

**Figure 2 polymers-14-01054-f002:**
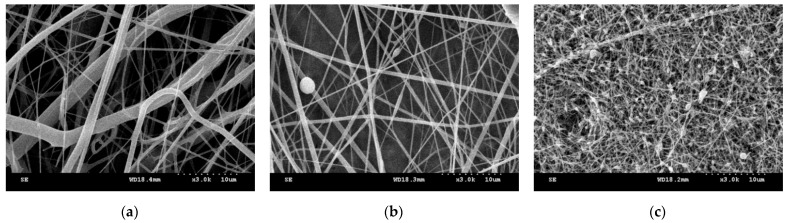
Fiber morphology of PVA nanofibrous membranes as related to a voltage being (**a**) 30 kV, (**b**) 50 kV, and (**c**) 70 kV.

**Figure 3 polymers-14-01054-f003:**
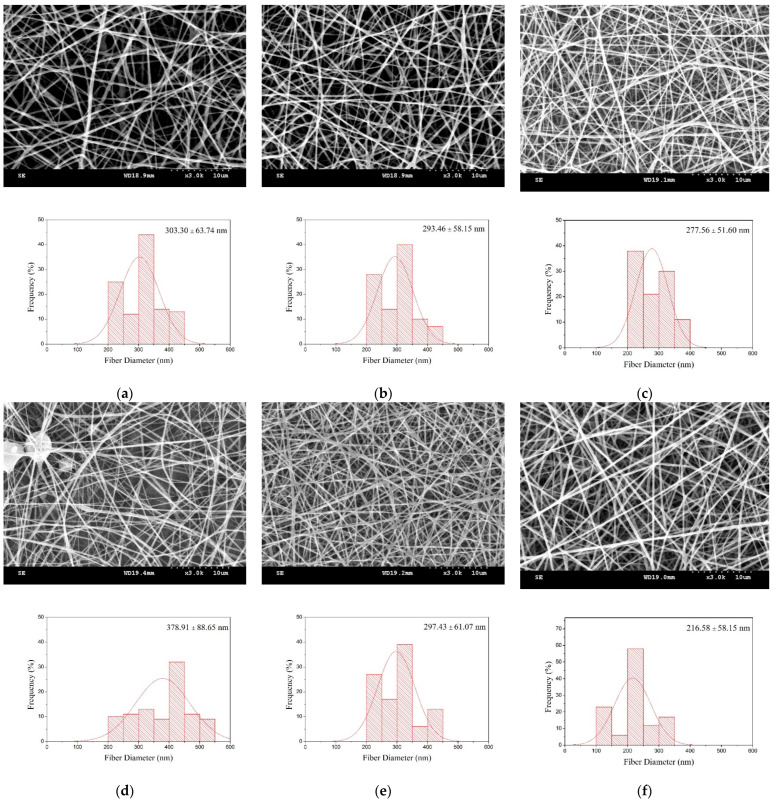
SEM images and diameter distribution of PVA/WS-CS nanofibrous membranes as related to blending ratio and WS-CS concentrations of (**a**) 80/20-05, (**b**) 80/20-10, (**c**) 80/20-15, (**d**) 60/40-05, (**e**) 60/40-10, and (**f**) 60/40-15.

**Figure 4 polymers-14-01054-f004:**
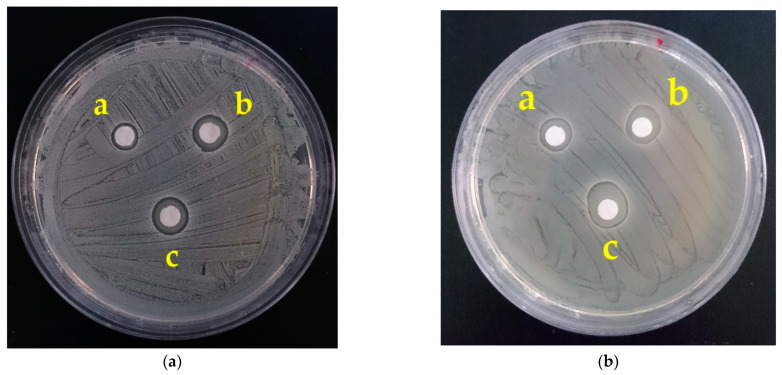
Bacteriostatic zone of PVA/WS-CS nanofibrous membranes with a blending ratio of (**a**) 80/20 and (**b**) 60/40, where the WS-CS concentrations were 5, 10, and 15 wt% for membranes a, b, and c, respectively.

**Figure 5 polymers-14-01054-f005:**
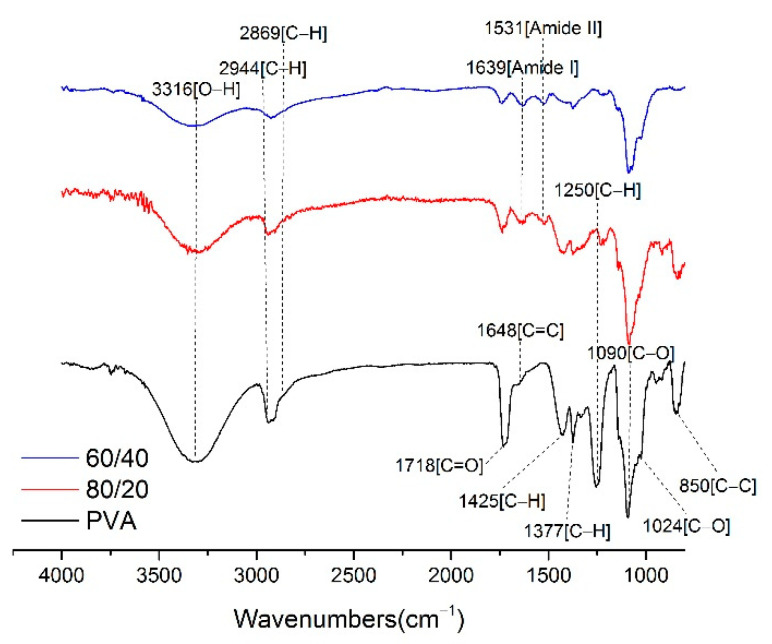
FTIR spectra of PVA/WS-CS nanofibrous membranes.

**Figure 6 polymers-14-01054-f006:**
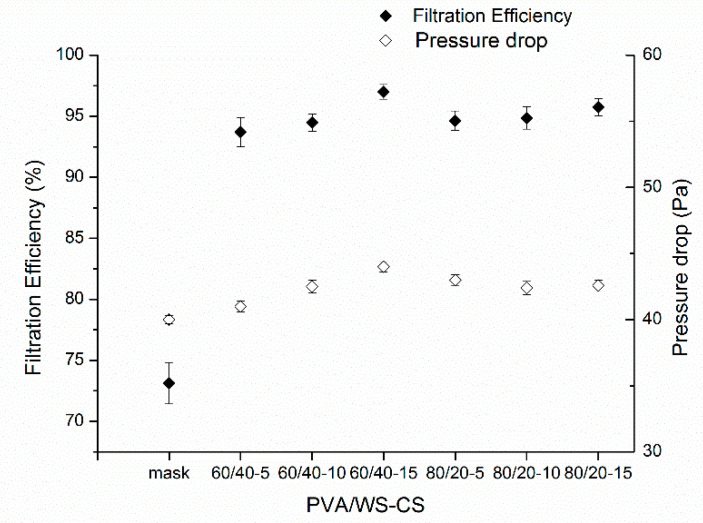
Filtration efficiency and pressure drop of PVA/WS-CS nanofibrous membranes.

**Figure 7 polymers-14-01054-f007:**
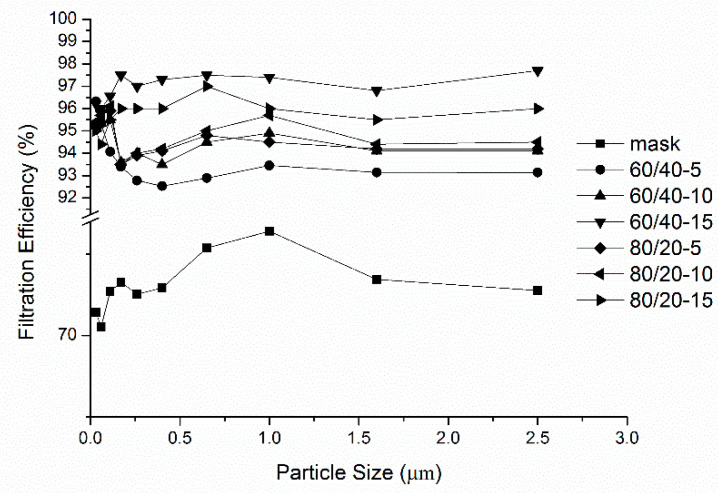
Filtration efficiency of PVA/WS-CS nanofibrous membranes as related to the particle size.

**Figure 8 polymers-14-01054-f008:**
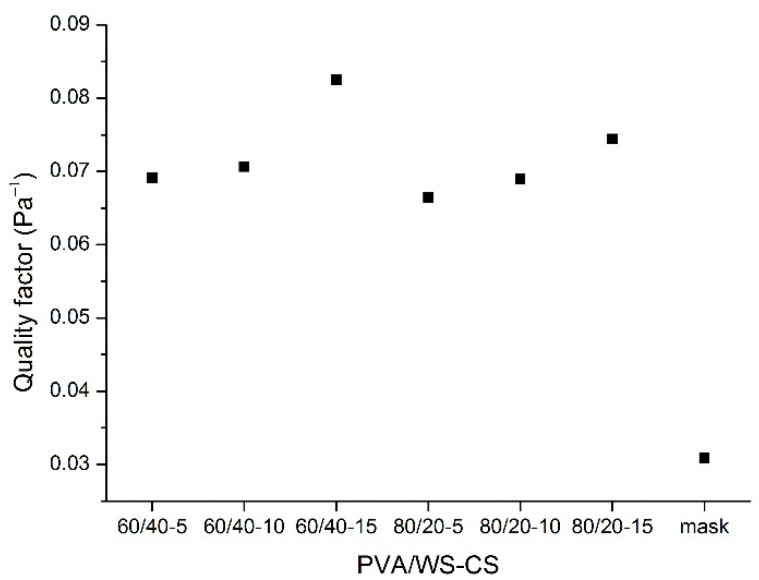
Quality factor of PVA/WS-CS nanofibrous membranes.

**Table 1 polymers-14-01054-t001:** Specifications of PVA/WS-CS mixtures and nanofibers.

Sample	Viscosity (cP)	Conductivity (μS/cm)	Diameter of Nanofibers (nm)
Pure PVA	584.1	0.67 × 10^3^	233.92 ± 30.77
PVA/WS-CS (80/20-05)	709.0	2.01 × 10^6^	303.30 ± 63.74
PVA/WS-CS (80/20-10)	792.7	2.97 × 10^6^	293.46 ± 58.15
PVA/WS-CS (80/20-15)	911.1	3.90 × 10^6^	277.56 ± 51.60
PVA/WS-CS (60/40-05)	788.4	3.44 × 10^6^	378.91 ± 88.65
PVA/WS-CS (60/40-10)	891.1	6.51 × 10^6^	297.43 ± 61.07
PVA/WS-CS (60/40-15)	991.6	8.07 × 10^6^	216.58 ± 58.15

**Table 2 polymers-14-01054-t002:** Characterizations of PVA/WS-CS nanofibrous membranes.

Sample	Pore Volume (cm³/g)	Pore Size (nm)	WVTR (g/(day × m^2^)
mask	0.009621	2824.07	2171.53 ± 54.18
PVA/WS-CS (80/20-05)	0.014591	19.73	1561.45 ± 31.65
PVA/WS-CS (80/20-10)	0.015361	18.91	1567.96 ± 24.78
PVA/WS-CS (80/20-15)	0.014436	17.02	1542.78 ± 35.18
PVA/WS-CS (60/40-05)	0.019788	22.48	1608.12 ± 45.16
PVA/WS-CS (60/40-10)	0.020176	17.93	1384.92 ± 29.92
PVA/WS-CS (60/40-15)	0.012706	12.06	1319.27 ± 30.88

## Data Availability

All data relevant to the study are included in the article.

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
