# Peer review of "Preparation of Needleless Electrospinning Polyvinyl Alcohol/Water-Soluble Chitosan Nanofibrous Membranes: Antibacterial Property and Filter Efficiency"

_polymers, 2022, doi:10.3390/polym14051054_

Round 1

Reviewer 1 Report

The manuscript "Preparation of Needleless Electrospinning Polyvinyl Alcohol/Water-Soluble Chitosan Nanofibrous Membranes: Antibacterial Property and Filter Efficiency" describes the preparation of filtering materials and their performance in mechanical particles removal and brocade activities.  The results is interesting,  but following comments have to addressed:
(i) Water permeability, porosity and pore size have to evaluate and discusse.  It's a key parameters of filtering materials. 
(ii) Ceramic membranes with biocide properties are widely used for microfiltration processes (doi.org/10.1016/j.ceramint.2014.04.080 ). It should be mentioned in Introduction section. 
(iii) Regeneration and membrane-fouling studies are important for real practical applications of prepered filtering materials.  Please, add information to the results and duscussion section. 
(iv) Biodegradable properties of chitosan based metareials coub be useful for utilizing the spent filtering materials (doi.org/10.1016/j.carbpol.2019.01.045). 

Author Response

Response to Reviewer 1 Comments

(i) Water permeability, porosity and pore size have to evaluate and discusse.  It's a key parameters of filtering materials. 

Answer: Thank you for your comment. The measurements and discussion regarding porosity, pore size, and WVTR have now been appended to the text in sections 2.5, 2.6, and 3.2. As the materials used in this study are all water soluble without particular cross-linking, the water permeability may not be suitable test item for the proposed nanofibrous membranes. Instead, we conduct the water vapor transmission rate measurement.

(ii) Ceramic membranes with biocide properties are widely used for microfiltration processes (doi.org/10.1016/j.ceramint.2014.04.080 ). It should be mentioned in Introduction section. 

Answer: Thank you for providing us the related literatures. The information about ceramic membranes has now been incorporated with the introduction.

(iii) Regeneration and membrane-fouling studies are important for real practical applications of prepered filtering materials.  Please, add information to the results and duscussion section. 

Answer: Thank you for your comment. Differing from the common membranes that emphasize the repetitive use, the PVA/WS-CS nanofibrous membranes are a one-off product and therefore repetitive types of tests are not ideal for the material. Containing a low-pollution process and natural materials as matrices, the proposed membranes are made just like the common membranes. Your suggestion enlightens and reminds us of the importance of regeneration and membrane fouling, and we will bear it in mind in the design of our future study and have appended related information to the introduction. Thank you.

(iv) Biodegradable properties of chitosan based metareials coub be useful for utilizing the spent filtering materials (doi.org/10.1016/j.carbpol.2019.01.045). 

Answer: Thank you for providing us the useful and related studies, which have been supplemented to our literature.

Reviewer 2 Report

This work presents electrospun nanofibers composed of PVA blended with water-soluble chitosan using needless electrospinning. The paper is very-well written with systematic analysis. I recommend publishing this work after minor revision, as follows:

1- More features about the advantages of needless electrospinning compared to the needle-technique should be discussed in the introduction.

2- Is the setup used Nanospider-Elmarco or hand-made setup in the lab? If Nanospider, you have to mention the full brand with model.

3- Why did you work with 50 kV? Did you try other HV values and what is the impact on morphology of fibers?

4- Given that you are targeting filtration, you have to add porosity analysis of all of your samples.

5- The relation between voltage drop and life of mask has to be properly cited.

6- A quick comparison between your results and other similar work in literature should be discussed.

7- The water solubility of your nanofibers could be a problem related to durability use as a mask. You need to cite similar compositions used for such applications along with the advantages over other non-soluble mats.

Author Response

Response to Reviewer 2 Comments

1- More features about the advantages of needless electrospinning compared to the needle-technique should be discussed in the introduction.

Answer: Thank you for your comment. The comparison between the needless electrospinning and the needle-technique has now been provided with the introduction.

2- Is the setup used Nanospider-Elmarco or hand-made setup in the lab? If Nanospider, you have to mention the full brand with model.

Answer: All of the nanofibrous membranes in this study are prepared using a self-made machine, the related description has now been incorporated with section 2.2. Thank you.

3- Why did you work with 50 kV? Did you try other HV values and what is the impact on morphology of fibers?

Answer: The pilot experiment has now been described and accompanied with images in section 3.1. The optimal voltage is determined according to the pilot experiment.

4- Given that you are targeting filtration, you have to add porosity analysis of all of your samples.

Answer: Thank you for your suggestion. Regarding the porosity, pore size, and water vapor transmission rate (WVTR), the measurements as well as the results and discussion have now been added to sections 2.5, 2.6, and 3.2.

5- The relation between voltage drop and life of mask has to be properly cited.

Answer: Thank you for your comment. The supportive literatures of pressure drop as well as the life of masks have now been incorporated with section 3.5.

6- A quick comparison between your results and other similar work in literature should be discussed.

Answer: Thank you for your comment. More literatures that are pertinent to PVA/CS nanofibrous membranes have now been compared in and appended to sections 3.2, 3.3, and 3.5.

7- The water solubility of your nanofibers could be a problem related to durability use as a mask. You need to cite similar compositions used for such applications along with the advantages over other non-soluble mats.

Answer: Thank you for your comment. A majority of literatures used PVA/CS filters for water filtration and metal ion adsorption filtration, which involved with related cross-linking treatment. Moreover, the air filters can be divided into one-off type and repetitive use type, the latter of which demands cross-linking. In other words, the conduction of cross-linking is accompanied with the pollution caused by cross-linking agents or a manufacturing process that is unfriendly to the eco-system. This study proposes one-off air filters with a water solution exclusively, using a low-pollution process. The related information and comparison have now been supplemented to the text.

Round 2

Reviewer 1 Report

The revised manuscript could be recommended for publication.